# Gender Differences Regarding Self-Perceived Physical and Mental Health in Spanish University Sports and Physical Therapy Students after Termination of the COVID-19 Lockdown Period

**DOI:** 10.3390/healthcare12020191

**Published:** 2024-01-12

**Authors:** Ismael García-Campanario, María Jesús Viñolo Gil, Luc E. Vanlinthout, Carlos Pérez Pérez, Cristina O’Ferrall González

**Affiliations:** 1Grupo PAIDI UCA CTS391, Department of Medicine, Faculty of Medicine, University of Cadiz, 11003 Cadiz, Spain; carlos.perez@uca.es; 2Department of Nursing and Physiotherapy, Institute for Biomedical Research and Innovation of Cádiz, University of Cadiz, 11009 Cadiz, Spain; mariajesus.vinolo@uca.es (M.J.V.G.); cristina.oferrall@uca.es (C.O.G.); 3Faculty of Medicine, University Hospital Gasthuisberg, University of Leuven, 3000 Leuven, Belgium; vanlinthout.l@skynet.be

**Keywords:** gender differences, mental health, university students, COVID-19, quality of life

## Abstract

The COVID-19 pandemic was an unprecedented situation that raised concerns about the physical and mental health of adolescents. Several surveys demonstrated that post-lockdown, women reported more complaints and lower perceived quality of life compared to men. The aim of this study was to analyze gender differences in self-reported physical and mental health immediately after the second lock-down restrictions (July 2020 to December 2020) were suspended and physical exercise classes resumed after a break of several months. This was achieved using a comparative cross-sectional survey of over-18-year-old students from the faculties of Sports Science and Physical Therapy at the University of Cadiz (UCA) in Spain. Quality of life was assessed using two types of questionnaires. The first covered quality of nutrition (PREDIMED), and the second assessed emotional impact (SF12). Physical activity level was estimated using the International Physical Activity Questionnaire (IPAQ). Of the 166 participants in this study, about two-thirds were men. Men had a better perception of their overall health quality than women. In addition, men had significantly fewer limitations in performing activities of daily living than their female counterparts. In contrast, female university students had better coping strategies, that is, they were better able to handle the problems of daily life and did so with more composure, attention, and concentration. These findings highlight the differences in post-release recovery between men and women and can be used to develop programs to promote better living standards and services to reduce gender disparities, which can ultimately improve quality of life.

## 1. Introduction

The first COVID-19 cases detected on European soil were reported in France as early as 24 January 2020. On 21 February, only 47 cases were registered in nine European countries; most of them were either linked to two clusters, in Bavaria (Germany) and Haute-Savoie (France), or related to a trip from China. The situation rapidly deteriorated in early March, especially after an outbreak in Northern Italy. On 11 March, the WHO declared a global pandemic, and following this, daily updates on new confirmed COVID-19 cases and deaths made the headlines all over the world. During the next 3 months, the spread and the fatality rate of the new virus took most countries by surprise [1].

On 14 March 2020, the Spanish government declared a state of emergency throughout Spain in response to the health crisis of COVID-19. To address the challenges posed by the pandemic, various social restrictions were implemented. These measures encompassed social distancing; limited travel; and regulated access to sports facilities, healthcare centers, universities, workplaces, and recreational venues. The consequences of social isolation and loneliness during the lockdown were linked to increased inactivity, sedentary lifestyles, weight gain, addictive behaviors, and mental health disorders, as well as inadequate exposure to sunlight [2]. A recent meta-analysis of longitudinal studies has shown that there was an increase in mental health symptoms during the first wave of the COVID-19 pandemic (March–April 2020), followed by a decline in symptoms to pre-pandemic levels by the summer of 2020 [3,4,5,6,7,8].

The analysis of data from the UK Household Longitudinal Study (UKHLS) has tracked changes in the levels of psychological distress during the pandemic. It suggests that the proportion of adults aged 18 years and above reporting a clinically significant level of psychological distress increased from 20.8% in 2019 to 29.5% in April 2020, before falling again to 21.3% in September 2020. There was a subsequent rise to 27.1% in January 2021, followed by a further decline to 24.5% by the end of March 2021. The prevalence of anxiety and depression was confirmed by the World Health Organization (WHO). The data suggest that the highest levels of depression and anxiety occurred in the early stages of lockdown but declined fairly quickly thereafter, possibly because the individuals were adapting to the conditions [9].

Adolescents are at a unique period in their lives when their social environment is important for crucial functions in brain development, self-concept construction, and mental health. Physical distancing might have a disproportionate effect on an age group for whom peer interaction is a vital aspect of development. Quarantine and isolation may also lead to acute stress disorder [10]. Mandatory confinement has forced university students to conform to online education, which has worsened their quality of life and mental health. The results of a survey in France suggest a high prevalence of mental health issues (suicidal thoughts, severe distress, a high level of perceived stress, severe depression, and a high level of anxiety) among students who experienced quarantine, underlining the need to reinforce prevention, surveillance, and access to care [11]. Adolescents may suffer from a range of psychological issues such as anxiety, fear, worry, depression, difficulty sleeping, and loss of appetite [12]. In Italy, a study was conducted with 655 university students who reported feelings of sadness (51.3%), nervousness (64.6%), irritability (57%), difficulty concentrating (55.9%), sleep problems (54.5%), eating disorders (73.6%), tachycardia (65%), or a tendency to cry (65%) [13]. After the removal of confinement restrictions, university students with no history of mental disorders were found to have a higher incidence of anxiety, depression, and insomnia, as well as lower levels of resilience [14].

The few studies that have focused on the health of university students during COVID-19 in Europe have found gender differences. A French study reported poorer mental health among women than men [15]. Spanish research showed more anxiety and loneliness among women than men [16]. On the other hand, no gender difference in anxiety was observed among students during the pandemic in China [17]. Meda et al. investigated students’ mental health before, during, and after the COVID-19 lockdown in Italy and reported that students experienced more severe depressive symptoms during the lockdown, which were independent of gender [18]. However, a recent systematic review and meta-analysis found that women had more symptoms of depression and anxiety than men [19,20]. Men were found to adopt healthier lifestyles than women during the COVID-19 pandemic, with more physical activity and better stress management. However, women acquired a greater capacity for interpersonal relationships. Nevertheless, female physical therapy students rated their health status and quality of life lower. They mentioned more signs and symptoms of depression, anxiety, stress, experiential avoidance, psychological inflexibility, inferior sleep quality, and loneliness than their male counterparts [21]. 

The purpose of this study was to analyze sex differences in self-reported physical and mental health after restrictions due to the pandemic were suspended and physical classes resumed after a break of several months. This was achieved using the SF12, PREDIMEC, and IPAQ physical activity questionnaires in a sample of sports science and physiotherapy students at the University of Cadiz, Spain. Our null hypothesis was that there were no significant differences between (i) genders and (ii) degrees in self-reported physical and mental health after the suspension of restrictions due to the pandemic.

## 2. Materials and Methods

This comparative cross-sectional study included over-18-year-old students of both biological sexes from the Departments of Sports Science and Physical Therapy at the University of Cadiz (UCA) immediately after the restrictions of the second lockdown were suspended (from July 2020 to December 2021). We focused on university sports and physical therapy students because they represent a homogeneous group within the university population. We did not consider individuals recovering from serious injuries or surgeries that might have affected their participation in sports or physical therapy activities. We also excluded pregnant individuals because of possible risks or limitations associated with certain exercises and activities during pregnancy. 

Within a quantitative survey design, determining the sample size is essential. One of the real advantages of quantitative methods is their ability to use smaller groups of people to make inferences about larger groups that would be prohibitively expensive to study [22]. The question is then how large a sample is required to infer research findings regarding a population.

The Sports Science and Physical Therapy Departments of the University of Cadiz (UCA) enroll about 300 students each year. It is estimated that 60% of university students were inadequately active during the COVID-19 pandemic [23]. The sample size was calculated using Cochran’s formula [24]. Taking this rate (60% inadequately active) as a reference, considering the number of students enrolled (*n* = 300), and assuming a confidence level of 95% and an estimation error of 5%, at least 166 students were needed for this study.

Through an online survey, sociodemographic and COVID-19-related information was collected from participants, such as whether he/she had suffered from a COVID-19-related illness in the past 6 months (yes/no); had symptoms such as fever, cough, loss of taste, loss of smell, respiratory problems; or required hospitalization (yes/no).

The International Physical Activity Questionnaire (IPAQ) is a widely used tool for assessing physical activity levels in various populations, including university populations [25]. It was developed by an international group of researchers to provide a standardized method for measuring physical activity. The IPAQ questionnaire consists of 7 items that assess the frequency, duration, and intensity of physical activity performed in the last week. Weekly activity is recorded in Mets (metabolic task equivalent or metabolic index units) per minute per week. IPAQ 1 (intensive activities): days per week that one has engaged in intense physical activity for more than 10 min at a time. IPAQ 2 (intense activities): hours/minutes per day of intense activity. IPAQ 3 (moderate activities): days per week of moderate physical activity for more than 10 min at a time. IPAQ 4 (moderate activities): hours/minutes per day of moderate activity. IPAQ 5 (low activity): days per week that one has been physically active, such as walking or visiting a place of leisure, for more than 10 min at a time. IPAQ 6 (low activity): hours/minutes per day of low-intensity physical activity. IPAQ 7: hours/minutes per day spent sitting. IPAQ 8: does not exist (the global result of the Mets).

Quality of life was assessed using two types of questionnaires. The first was related to diet quality and the second assessed emotional impact. 

Diet quality was determined by adherence to the Mediterranean diet, which is associated with various health benefits. For this purpose, the Prevention with Mediterranean Diet (PREDIMED) questionnaire was used, which is composed of 14 dichotomous (1/0) items [26] (see Table 1.). These dichotomous questions simplify the response process and make the questionnaire relatively quick and straightforward for participants to complete. This instrument has been validated for the university population [27]. A score below 9 points means that nutritional quality needs to be improved. While it may not be a direct measurement of overall quality of life, it can provide valuable information about the influence of dietary patterns on health-related outcomes, which in turn can affect quality of life.

The emotional impact of health on the daily lives of people older than 14 years was assessed using the 12-item Short Form Survey (SF-12) [28]. The SF-12 is a shorter version of the SF-36. It provides a snapshot of health-related quality of life, making it easier for individuals to complete and for researchers or health professionals to analyze.

The SF-12 uses eight domains: (I) general health (GH), assessed with item 1, which includes the personal assessment of health; (II) physical functioning (PF) (items 2 and 3), which evaluates the degree to which health limits physical activities such as climbing stairs or walking for more than one hour; (III) role-physical (RP) (items 4 and 5), determining the degree to which physical health affects work and other daily activities; (IV) role-emotional (RE) (items 6 and 7), appraising the degree to which emotional problems affect work or daily activities; (V) bodily pain (BP) (item 8), estimating the effect of pain on work and home life; (VI) mental health (MH) (items 9 and 11), gauging feelings of calmness, peacefulness, discouragement, and sadness; (VII) energy/fatigue (VT) (item 10), measuring the degree of vitality versus that of fatigue and exhaustion; and (VIII) social function (SF) (item 12), rating the degree to which physical or emotional health problems interfere with daily life. Items are rated on a Likert scale with scores ranging from 1 to 6 points, resulting in total scores ranging from 12 to 47 points. The higher the score, the better the health-related quality of life. This instrument has been validated in the general population [29,30,31]. To obtain information on the level of well-being and the functional capacity of people older than 14 years, the SF-12 health-related quality of life questionnaire was used [28].

Google Forms was used to create online forms and surveys with multiple question types. The form was sent to the students through the class delegates. The data were exported to Excel, where the quality control of the information and coding was carried out. The results were analyzed in real time. Participation was completely voluntary and anonymous. The study was conducted in accordance with the ethical standards for human experimentation according to the Declaration of Helsinki (www.cirp.org/library/ethics/helsinki (accessed on 15 March 2020)) and was approved by the Bioethics Committee of the University of Cadiz (Reference number: 164 008/2021).

### Statistical Analysis of the Data

The comparative analysis of the survey data was performed using frequencies and percentages for qualitative variables and means and standard deviations (SDs) for quantitative variables. Several hypothesis tests were then performed, based on the partial and final scores of each of the questionnaires used. After confirming normality using Kolmogorov–Smirnov, Shapiro–Wilk, or Q–Q plots, Student’s *t*-test was used. If the assumption of normality was not met, equivalent nonparametric tests were applied, such as the Mann–Whitney U-test or the Wilcoxon rank sum test. 

To relate a dependent binary variable (yes/no or 1/0) to a set of independent variables concerning gender (male/female) or degree (sports/physical therapy), we used logistic regression. To relate ordinal dependent variables to a set of independent variables (male/female or sports/physical therapy), we used ordered logistic regression. Ordered logistic regression is an extension of logistic regression that is used when the dependent variable is ordinal, meaning that it has more than two ordered categories but the categories have a meaningful order, e.g., “never”, “sometimes”, “many times”, “always”.

A significance level of 5% was applied, and IBM SPSS v.26 was used.

## 3. Results

Between 1 January and 31 March 2022, we included a total of 303 students, with 63.2% being male. Out of this group, 133 (44.19%) were pursuing a degree in sports science, while the remaining 168 (55.81%) students were studying physical therapy. The average age of the entire group was 21.48 (4.57) years. In terms of gender-specific averages, men had a mean age of 21.8 (5.34) years, and women had a mean age of 20.93 (2.73) years, with no statistically significant differences. Notably, there was a significantly higher percentage of male participants in the physical activity and sports science program (76.78%) compared to those in the physiotherapy program (46.27%). Thirty-two percent had experienced COVID-19, but none of them required hospitalization. The most prevalent symptoms reported were fever and cough. No significant differences based on gender were observed (see Table 2).

In our samples, all the parameters studied had a normal distribution, which allowed us to use the *t*-test, which is a powerful statistical tool.

Regarding health-related quality of life, assessed with the SF-12 questionnaire, significant sex differences were found in 9/12 items (SF1, SF3, SF4, and SF7-SF12).

Men believed that they had a greater perception of their state of health. Among the physiotherapy and physical activity students, we also found significant differences in health limiting some activities of daily life (SF3 (physical function) and SF10 (energy/fatigue)). In this case, men reported having fewer limitations than women in performing activities of daily life due to their health status. We also found significant differences between both degrees in relation to levels of anxiety and depressive states, with women tending to cope better with various emotional conditions than men. Regarding physical pain, we found significant differences between university degrees and genders. As far as mental health is concerned, we found significant differences between both degrees and gender, with men showing better mental health despite managing these states more poorly than women. We also found significant differences between degrees and gender in social functioning. (see Table 3).

There were significant differences in adherence to the Mediterranean diet between men and women. There were significant sex differences with respect to PREDIMED item 2 (consumption of olive oil); PREDIMED item 5 (consumption of red meat, hamburgers, or meat products (ham, sausage, etc.)); and PREDIMED item 13 (consumption of chicken, turkey, or rabbit meat instead of veal, pork, hamburgers, or sausage). Women, compared to men, consumed less olive oil and greater amounts of red meat or processed meat and chicken, turkey, or rabbit. Students in our study adhered poorly to the Mediterranean diet (about one-third). There were no differences between men and women in the degree of adherence to the Mediterranean diet (see Table 4.).

Regarding the IPAQ, there were significant differences between men and women for both university degrees in terms of vigorous and moderate physical activities (IPAQ1–IPAQ3 (vigorous–moderate physical activities)). Women recorded more physical activities of a low or moderate intensity than men. It is noteworthy that students of sports science, due to their academic curriculum, met higher weekly recommendations for healthier practice through physical exercise (see Table 5).

The null hypothesis that there were no significant differences between genders and degrees in self-reported physical and mental health after the suspension of restrictions due to the pandemic could be rejected.

## 4. Discussion

The present study examined the extent to which the COVID-19 pandemic affected the physical and mental health of university sports and physical therapy students. Significant sex differences regarding self-perceived quality of life became apparent. Men had a better perception of their overall health than women. Moreover, men had significantly fewer limitations in performing activities of daily life than their female counterparts. In contrast, women’s emotional roles were significantly different from those of men. Female sports and physical therapy students had better coping strategies, i.e., they were able to handle the problems of daily life more successfully and did so with more temperance, attention, and concentration. 

Previous studies showed lower self-reported health quality of life in women than in men. Our results regarding the gender-specific effect of lockdown on physical and mental health differed from those of other studies. 

Even before the COVID-19 pandemic, US women had lower health-related quality of life scores than men [32]. In a sample of 125,732 11-to-15-year-olds from 29 European and North American countries, participating in a collaborative WHO study, girls reported poorer overall health than boys [33]. In a German study, self-reported general health in 10-to-31-year-olds was better in male than in female participants, whereas the decrease in self-reported general health with increasing age was more pronounced in girls [34]. The few studies that have focused on the health of university students during the COVID-19 pandemic have found differences between men and women. Men were found to be at lower risk of pandemic-related stress than women while being exposed to similar healthcare restrictions [15,16,35,36].

Men and women exhibit different biological, behavioral, social, and cultural characteristics. Sex-related differences are apparent in social interactions, lifestyles, health perceptions, and actions individuals take to influence their health status [37,38]. A recent systematic review and meta-analysis found that women had more symptoms of depression and anxiety than men [19,20]. The gender gap in feelings of distress could be explained by differences (1) in family and caring responsibilities; (2) in financial and work situations; (3) in social engagement; and (4) in health status and health behaviors, including physical activity.

The apparent contradiction between the better coping strategies among female sports and physical therapy students in this study and the higher stress levels among women in other studies could be explained by the higher physical activity levels among our students. Women who decreased their exercise due to limitations during the COVID-19 pandemic showed significantly lower showed lower mental health scores; decreased social, emotional, and psychological well-being; and significantly higher anxiety levels [39]. Women who reduced their physical activity perceived it as less enjoyable and experienced a greater number of barriers to physical activity than men [40,41].

The benefits of physical activity are well known and extensively documented in the scientific literature. There is a positive relationship between physical activity and various indicators of quality of life. In addition to the apparent physical health benefits, physical activity also affects mental health positively. Physically inactive individuals have been reported to have higher rates of morbidity and healthcare expenditures and inferior self-perceived quality of life. Furthermore, thorough evaluations of global studies have discovered that a small amount of physical exercise is sufficient to provide health benefits. Moderate levels of physical activity were associated with a lower prevalence of COVID-19-related hospitalizations, and there was a positive relationship between physical activity and various indicators of quality of life [42,43,44,45,46,47].

Differences were found between sports and physiotherapy students regarding SF3 (physical function) and SF10 (energy/fatigue). Also, in this case, men reported having fewer limitations than women in performing activities of daily life due to their health status. The apparent difference in physical condition between these two groups of students may be explained by the differences in their training programs. Sports students may engage in more intense and regular physical activity as part of their training, which could contribute to higher fitness levels compared to physiotherapy students., While being knowledgeable about rehabilitation and exercise, physiotherapy students may not necessarily engage in the same level of sport-specific training. Sports students typically focus on specific sports or athletic activities, while physiotherapy students may be exposed to a broader range of exercises and therapeutic interventions.

Some strengths and limitations of the current study must be considered. We focused on university sports and physical therapy students because they are a homogeneous group within the university population. Sports and physical therapy programs attract students who have a specific interest in these areas. They do not necessarily represent the general student population. Quality of life is generally considered a qualitative parameter because it includes subjective elements related to an individual’s overall well-being, satisfaction, and happiness. Physical and mental health are interconnected aspects of overall well-being. However, researchers and healthcare professionals often use standardized instruments to try to measure quality of life more objectively/quantitatively, i.e., the SF-12 and PREDIMED questionnaires.

While these tools are valuable for quantitative assessments, it is essential to recognize that quality of life is a complex and multidimensional concept. It relates not only to physical health, but also to psychological, social, and environmental factors. Therefore, a comprehensive understanding of an individual’s quality of life often requires a combination of quantitative measures and qualitative assessments, such as interviews or open-ended questions, to capture the subjective aspects of well-being. The current study was an online survey; no interviews were conducted. 

Sex differences should be taken into account when addressing students’ mental health. The results of this study can be used to develop programs to promote better living standards and services to reduce gender disparities, which could ultimately improve quality of life. Tailoring programs and support services to meet the specific needs and interests of university sports and physical therapy students could contribute to a more enriching and targeted educational experience.

## 5. Conclusions

The present study examined gender differences regarding self-perceived physical and mental health in Spanish university sports and physical therapy students after termination of the second COVID-19 Lockdown Period (July 2020-December 2020). Men had a better perception of their general health than women. In addition, men had significantly fewer limitations in performing activities of daily living than their female counterparts. In contrast, female sports and physical therapy students had better coping strategies, i.e., they were able to handle daily life problems more successfully and did so with more temperance, attentiveness, and concentration. Previous studies showed lower self-reported health quality of life in women than in men. Our results on the gender-specific effect of confinement on physical and mental health differed from those of other studies. The apparent contradiction between better coping strategies among female sports and physical therapy students in this study and higher stress levels among females in other studies could be explained by the higher levels of physical activity among our female students. Women who decreased their exercise due to limitations during the COVID-19 pandemic showed significantly lower showed lower mental health scores; decreased social, emotional, and psychological well-being; and significantly higher anxiety levels then men.

## Figures and Tables

**Table 1 healthcare-12-00191-t001:** Validated 14-item questionnaire of Mediterranean diet adherence [26].

Questions	Criteria for 1 Point
1. Do you use olive oil as main culinary fat?	Yes
2. How much olive oil do you consume in a given day (including oil used for frying, salads, out-of-house meals, etc.)?	≥4 tbsp
3. How many vegetable servings do you consume per day? (1 serving: 200 g (consider side dishes as half a serving))	≥2 (≥1 portion raw or as a salad)
4. How many fruit units (including natural fruit juices) do you consume per day?	≥3
5. How many servings of red meat, hamburger, or meat products (ham, sausage, etc.) do you consume per day? (1 serving: 100–150 g)	<1
6. How many servings of butter, margarine, or cream do you consume per day? (1 serving: 12 g)	<1
7. How many sweet or carbonated beverages do you drink per day?	<1
8. How much wine do you drink per week?	≥7 glasses
9. How many servings of legumes do you consume per week? (1 serving: 150 g)	≥3
10. How many servings of fish or shellfish do you consume per week? (1 serving: 100–150 g of fish or 4–5 units or 200 g of shellfish)	≥3
11. How many times per week do you consume commercial sweets or pastries (not homemade), such as cakes, cookies, biscuits, or custard?	<3
12. How many servings of nuts (including peanuts) do you consume peer week? (1 serving: 30 g)	≥3
13. Do you preferentially consume chicken, turkey, or rabbit meat instead of veal, pork, hamburger, or sausage?	Yes
14. How many times per week do you consume vegetables, pasta, rice, or other dishes seasoned with sofrito (sauce made with tomato and onion, leek, or garlic and simmered with olive oil)?	≥2

Short questionnaire to assess adherence to the Mediterrean diet. The PREDIMED (in Spanish: PREvención con DIeta MEDiterránea) questionnaire is a 14-item instrument developed in a Spanish case-control study of myocardial infarction [26], in which the best cut-off points were selected for each food or food group to distinguish between cases and controls. The baseline 14-item questionnaire was the primary measure used in this study to appraise adherence of participants to the Mediterranean diet.

**Table 2 healthcare-12-00191-t002:** Descriptive analysis of the sample.

		Physiotherapy Degree			Sports Degree		
		Total	Men	Women	Total	Men	Women
		*n* (%)	*n* (%)	*n* (%)	*n* (%)	*n* (%)	*n* (%)
Course	1°	47 (35.3)	24 (51.1)	23 (48.9)	71 (42.0)	47 (66.2)	24 (33.8)
	2°	30 (22.6)	12 (40.0)	18 (60.0)	27 (16.0)	26 (100)	0 (0)
	3°	40 (30.1)	18 (45.0)	22 (55.0)	41 (24.3)	32 (78)	9 (22)
	4°	16 (12.0)	7 (43.8)	9 (56.3)	29 (17.2)	23 (79.3)	6 (20.7)
BMI	Underweight	9 (6.8)	2 (22.2)	7 (77.8)	4 (2.4)	2 (50)	2 (50)
	Normal weight	90 (67.7)	38 (42.2)	52 (57.8)	130 (77.4)	93 (72.1)	36 (27.9)
	Overweight	27 (20.3)	16 (59.3)	11 (40.7)	29 (17.3)	29 (100)	0 (0)
	Moderate obesity	6 (4.5)	4 (66.7)	2 (33.3)	3 (1.8)	3 (100)	0 (0)
	Severe obesity	1 (0.8)	1 (100)	0 (0)	2 (1.2)	2 (100)	0 (0)
Have you had COVID-19 in the last 6 months?	Yes	39 (29.1)	12 (30.8)	27 (69.2)	58 (34.3)	45 (77.6)	13 (22.4)
	No	95 (70.9)	50 (52.6)	45 (47.4)	111 (65.7)	84 (76.4)	26 (23.6)

**Table 3 healthcare-12-00191-t003:** Health-related quality of life, assessed with the 12 item Short Form Health Survey 12 (SF-12).

Item	*n* (%) Women	*n* (%) Men	*p* Univariable Regression Women vs. Men	*p* Multivariable Regression Women vs. Men ^‡^ Sport vs. Physiotherapy^¶^
**1. SF1: In general, would you say your health is:**			0.039 *	0.092
- excellent	3 (2.65%)	26 (13.68%)		0.360
- very good	55 (48.67%)	90 (47.37%)		
- good	49 (43.36%)	65 (34.21%)		
- regular	5 (4.42%)	9 (4.74%)		
- bad	1 (0.88%)	0 (0%)		
**2. SF2: Moderate activities, i.e., bowling or playing golf:**			0.071	0.044 ^‡,^*
- no, it does not limit me at all	105 (92.11%)	184 (96.84%)		0.354 ^¶^
- yes, it limits me a bit	7 (6.14%)	6 (3.16%)		
- yes, it limits me a lot	2 (1.75%)	0 (0%)		
**3. SF3: Climbing several flights of stairs:**			<0.0001 *	0.001 ^‡,^*
- no, it does not limit me at all	84 (73.68%)	174 (91.58%)		0.018 ^¶,^*
- yes, it limits me a bit	7 (23.68%)	16 (8.42%)		
- yes, it limits me a lot	3 (2.63%)	0 (0%)		
**4. SF4: Accomplished less than you would like:**			0.0007 *	0.287 ^‡^
- yes	24 (21.05%)	30 (15.79%)		0.907 ^¶^
- no	90 (78.95%)	160 (84.21%)		
**5. SF5: Were limited in the KIND of work or other activities:**			0.0744	0.035 ^‡,^*
- yes	21 (18.42%)	21 (11.11%)		0.168 ^¶^
- no	93 (81.58%)	168 (88.89%)		
**6. SF6: Accomplished less than you would like:**			0.8389	<0.0001 ^‡,^*
- yes	69 (60.53%)	72 (38.30%)		0.912 ^¶^
- no	45 (39.47%)	116 (61.70%)		
**7. SF7: Didn’t do work or other activities as carefully as usual:**				
- yes	58 (49.12%)	65 (34.39%)	0.0046 *	
- no	56 (49.12%)	124 (65.61%)		
**8. SF8: How much did PAIN interfere with your normal work?**			0.020 *	0.006 ^‡,^*
- nothing	68 (60.71%)	140 (74.47%)		0.097 ^¶^
- a bit	28 (25.00%)	27 (14.36%)		
- regular	10 (8.93%)	15 (7.98%)		
- fairly	5 (4.46%)	6 (3.19%)		
- a lot	1 (0.89%)	0 (0%)		
**9. SF9: Have you felt calm and peaceful?**			<0.0001 *	<0.0001 ^‡,^*
- never		3 (1.59%)		0.110 ^¶^
- sometimes	43 (37.72%)	41 (21.69%)		
- only some of the time	17 (14.91%)	12 (6.35%)		
- many times	26 (22.81%)	60 (31.75%)		
- almost always	19 (16.67%)	64 (31.75%)		
- always	5 (4.39%)	9 (4.76%)		
**10. SF10: Did you have a lot of energy?**			0.001 *	0.019 ^‡,^*
- never	1 (0.88%)	1 (0.53%)		0.013 ^¶,^*
- sometimes	47 (41.59%)	56 (29.63%)		
- only some of the time	13 (11.50%)	11 (5.82%)		
- many times	27 (23.59%)	54 (28.57%)		
- almost always	23 (20.35%)	59 (31.22%)		
- always	2 (1.77%)	8 (4.23%)		
**11. SF11: Have you felt downhearted and blue?**			<0.0001 *	0.001 ^‡,^*
- never	6 (5.36%)	21 (11.11%)		0.799 ^¶^
- sometimes	43 (37.72%)	61 (32.28%)		
- only some of the time	25 (21.93%)	68 (35.98%)		
- many times	26 (22.81%)	26 (13.76%)		
- almost always	13 (11.40%)	12 (6.35%)		
- always	1 (0.88%)	1 (0.53%)		
**12. SF12: Physical/emotional problems interfered with social activities?**			0.030 *	0.040 ^‡,^*
- never	41 (35.96%)	94 (49.47%)		0.908 ^¶^
- sometimes	28 (24.56%)	39 (20.53%)		
- only some of the time	34 (29.82%)	45 (23.68%)		
- many times	0 (0%)	0 (0%)		
- almost always	9 (7.89%)	10 (5.26%)		
- always	2 (1.75%)	2 (1.05%)		

Health-related quality of life, assessed with the Short Form Health Survey 12 (SF-12) which consists of 12 questions covering physical and mental health domains. The SF-12 is a self-reported outcome measure assessing the impact of health on an individual’s quality of everyday life. In this table, women and men are separately assessed. *n* (%): numbers (percent of total) of the gender group considered (women or men). *: significantly different; *p* value of women versus men (¶) or sports’ versus physiotherapy students (‡) in multivariable regression.

**Table 4 healthcare-12-00191-t004:** PREDIMED questionnaire.

Item	Women	Men	*p* Women vs. Men
PREDIMED1			
- yes	107 (93.86%)	179 (93.72%)	0.9603
- no	7 (6.14%)	12 (6.28%)	
PREDIMED2			
- yes	29 (25.44%)	78 (40.63%)	0.0071 *
- no	85 (74.56%)	114 (59.38%)	
PREDIMED3			
- <than 2 servings	62 (54.39%)	116 (60.42%)	0.3011
- ≥2 servings	52 (45.61%)	76 (39.85%)	
PREDIMED4			
- yes	39 (34.51%)	66 (34.38%)	0.9804
- no	74 (65.49%)	126 (65.63%)	
PREDIMED5			
- yes	81 (71.05%)	103 (53.98%)	0.0031 *
- no	33 (28.95%)	88 (46.07%)	
PREDIMED6			
- yes	86 (75.44%)	142 (74.35%)	0.8316
- no	28 (24.56%)	49 (25.69%)	
PREDIMED7			
- yes	74 (64.91%)	121 (63.35%)	0.7835
- no	40 (64.91%)	70 (36.65%)	
PREDIMED8			
- yes	0 (0%)	1 (0.52%)	0.4390
- no	114 (100%)	190 (99.48%)	
PREDIMED9			
- yes	61 (53.51%)	114 (60.32%)	0.2451
- no	53 (46.49%)	75 (39.68%)	
PREDIMED10			
- yes	45 (39.47%)	66 (34.55%)	0.3877
- no	69 (60.53%)	125 (65.45%)	
PREDIMED11			
- less than twice/week	62 (54.39%)	119 (61.98%)	0.1914
- more than twice/week	52 (45.61%)	73 (38.02%)	
PREDIMED12			
- yes	48 (42.11%)	86 (45.26%)	0.5913
- no	66 (57.89%)	104 (54.74%)	
PREDIMED13			
- yes	89 (78.6%)	128 (67.37%)	0.0334 *
- no	24 (21.24%)	62 (32.63%)	
PREDIMED14			0.1070
- yes	79 (69.30%)	149 (77.60%)	
- no	35 (30.70%)	43 (22.40%)	
CONCLUSION			0.8499
PREDIMED score <9	73 (64.04%)	125 (65.10%)	
PREDIMED score ≥9	41 (35.96%)	67 (34.90%)	

Adherence to the Mediterranean diet, assessed with 14-item PREDIMED questionnaire. The PREDIMED instrument is a self-reported outcome measure that is valuable in both clinical and research settings because it provides a standardized way to quantify dietary quality based on proven healthy eating patterns. In this table, women and men are separately assessed. *n* (%): numbers (percentage of total) of the gender group considered (women or men). Low and high adherence to the Mediterranean diet correspond to PREDIMED scores of <9 and ≥9, respectively. *: significant difference between sexes.

**Table 5 healthcare-12-00191-t005:** IPAQ questionnaire.

Item	Women	Men	*p* Women vs. Men
Age	20.89 (2.70)	21.78 (5.33)	0.1014
BMI	21.88 (2.87)	23.93 (3.37)	<0.0001 *
IPAQ1	2.61 (1.88)	3.32 (1.82)	0.0012 *
IPAQ2	68.90 (43.78)	83.37 (49.75)	0.0151 *
IPAQ3	2.09 (1.85)	2.56 (2.05)	0.0462 *
IPAQ4	61.99 (72.40)	62.67 (53.38)	0.9295
IPAQ5	2.26 (2.00)	5.35 (2.05)	0.7158
IPAQ6	69.01 (52.01)	77.11 (75.87)	0.3308
IPAQ7	5.99 (1.88)	5.51 (1.88)	0.0749
IPAQ8	298.65 (210.59)	267.15 (248.55)	0.2752

The International Physical Activity Questionnaire (IPAQ) is a widely used tool for assessing physical activity levels in various populations. In this table, women and men are separately assessed. *: significant difference between the sexes.

## Data Availability

Data will be provided on request.

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
