# Peer review of "Gender Differences Regarding Self-Perceived Physical and Mental Health in Spanish University Sports and Physical Therapy Students after Termination of the COVID-19 Lockdown Period"

_healthcare, 2024, doi:10.3390/healthcare12020191_

Round 1

Reviewer 1 Report

Comments and Suggestions for Authors

The effects on people due to the COVID-19 pandemic and the lockdown measures to control the psychological effects of these measures is a source of a large volume of research into this phenomenon.  This research extends the knowledge of these effects within the overall pandemic research field.  The research is required to understand the coping behaviors of those affected by viral pandemics and to construct support mechanisms to support those affected.

The stated aim for this research was ‘s to analyze gender differences in self-reported physical and mental health after restrictions due to the pandemic were suspended and physical classes resumed after a break of several months.’  The research data was collected via two questionnaires; ‘quality of nutrition (PREDIMED) and the second assessed the emotional impact (SF12).’  These survey tools are appropriate to the aim of study by assessing the quality of nutrition (PRIDIMED) and the second assessed the emotional impact (SF12), a self-reported outcome measure assessing the impact of health on an individual’s everyday life.  The data collected from these two surveys was integrated to inform the respondents about quality of life.

The conclusions of the research indicate that: ‘Men were found to have a better perception of their overall health than women. Furthermore, males had significantly fewer limitations in performing activities of daily living than their female counterparts. Female university students…had better coping strategies, i.e. were able to handle the problems of everyday life more successfully, and did so with more temperance, attention and concentration...The apparent contradiction between the better coping strategies among female undergraduates in our study and the higher levels of distress among women in other studies could be explained by the higher levels of physical activity among our students.’

The Abstract provides a good precis of research with sufficient detail to inform others of the current research.  The Introduction relates prior research of the phenomenon concisely and constructs the basis for the research aims in an understandable manner.  The Methods and Materials details the research design and implementation of the surveys, and the statistical methods employed in their analysis.  The Methods and Materials are appropriate to the research and provide excellent data for the research. The Results of the analyzed data (accompanied by graphical presentations of the analyses data) are understandable, relevant, and appropriate for the research and informs the interpretative Discussion.  The Discussion provides plausible explanations of the analyzed data (Results).

There are a few suggestions to improve the English presentation of the paper:

Lines 25, 26: ‘This was done using a descriptive cross-sectional study including more than of 18-year- old students’

Line 268: ‘The current study assesses the extent’

Line 281: ‘COVID-19 in have found gender differences.’

A commendable research project that will add to the literature of post-COVID-19 research.

Comments on the Quality of English Language

There are a few suggestions to improve the English presentation of the paper:

Lines 25, 26: ‘This was done using a descriptive cross-sectional study including more than of 18-year- old students’

Line 268: ‘The current study assesses the extent’

Line 281: ‘COVID-19 in have found gender differences.’

Author Response

Estimado revisor, nos gustaría agradecerle por sus comentarios y valoración de nuestro artículo. Te enviamos tu sugerencia con los cambios. Atentamente.

Reviewer 2 Report

Comments and Suggestions for Authors

line 68 needs a references

There may be differences due to the study being in Spain. (and the questionnaires were based on a Mediterranean diet - cultural differences). Also, the focus of the title/abstract could be more on the population -  Sports Science and Physiotherapy students. could be clearer that these are being compared. 

In the method section, more detail could have been included about the participants, recruitment, and exclusion/inclusion criteria. How was gender defined (or was it biological sex you looked at?)? 

When was the survey run? 

The qualitative variables were not very clear what these were and how they were used. 

What were the hypotheses set? 

The table in the results needs to be checked. 

Table 3: SF-12 questionnaire on quality of life could have the questions listed.

The link between the questionnaires and the COVID-19-related information in the results was not clear. For example, being vaccinated and the SF-12.  

The format of the article needs to be checked. 

Line 277 needs some references to support it. Also, are these findings different from before covid 19? (has anything changed) .

The discussion could have included some consideration of the limitations of the study. Also, more detail about the findings could be included to further support the discussion. 

Comments on the Quality of English Language

Generally, the English Language was okay throughout the paper.

Author Response

Dear Reviewer: We would like to thank you for the recommendations and your important scientific contribution to our manuscript. Best regards.

Reviewer 3 Report

Comments and Suggestions for Authors

Please specify - in which countries did the rate increase in the first Corona wave (lines 41-42).

Correct the placement of the sources (line 46) or provide sources for the information in lines 46 and 47.

Clarification: what characteristics (mental stability?) are involved in relation to mental health? (line 67)

Please specify what the overall goal of the study is. Health-promoting programs tailored to target groups? (line 94)

Check the statement that the study is only descriptive against the background of the analyzes used (line 100)

When was the data collected? How long was the lockdown phase over? (line 111)

Specify what exactly you want to achieve by calculating the sample size (line 104)

Somewhat surprisingly, data on vaccination etc. was also collected - please explain the connections with the study (lines 108 - 109)

Are there statements about “good” or “bad values” in relation to the questionnaires and scores (lines 112-157) – add these.

Specify the form of the T-test that was used (line 177)

Surprisingly, the BMI was collected using the add method (line 215).

What role does data on vaccinations or COVID illnesses play in the results?

Also discuss the data with the collection period of your study - what could have already recovered?

What might take even longer?

What follow-up questions would be important to clarify further important questions about gender-specific

Author Response

(The authors gave the same response as above.)
